# Acute Cardiopulmonary Response of High-Intensity Interval Training with Elastic Resistance vs. High-Intensity Interval Training on a Treadmill in Healthy Adults

**DOI:** 10.3390/ijerph20126061

**Published:** 2023-06-06

**Authors:** Letícia Nascimento Santos Neves, Victor Hugo Gasparini-Neto, Richard Diego Leite, Luciana Carletti

**Affiliations:** Laboratory of Exercise Physiology (LAFEX), Physical Education and Sports Center, Federal University of Espírito Santo (CEFD-UFES), Vitória 29075-910, ES, Brazil

**Keywords:** high-intensity interval exercise, lactate, single session, psychophysiological, subjective

## Abstract

This study aims to describe and compare cardiopulmonary and subjective responses during high-intensity interval training with elastic resistance (EL-HIIT) and traditional high-intensity interval training (HIIT) sessions. Twenty-two healthy adults (27.6 ± 4.4 years) performed an EL-HIIT and a HIIT protocol consisting of 10 × 1 min at ~85% $\overset{\cdot }{V}O_{\mathrm{2max}}$ prescribed by cardiopulmonary-specific tests. Pulmonary parameters, heart rate (HR), blood lactate, and rate of perceived exertion (RPE) were measured during exercise. Paired *t*-test and Cohen’s *d* effect size were used to compare peak and average values. Two-way repeated measures ANOVA and mixed model with Bonferroni’s post hoc test were used to compare each bout during the session. The EL-HIIT session showed higher peak and average values of HR, ventilation ($\overset{\cdot }{V}E$), relative and absolute oxygen uptake ($\overset{\cdot }{V}O_{2}$), carbon dioxide production ($\overset{\cdot }{V}CO_{2}$), and RPE than HIIT (*p* < 0.05). The effect size (ES) comparing HIIT and EL-HIIT was large for HR, $\overset{\cdot }{V}E$, and lactate (≥0.8) in peak values. Comparing each bout between HIIT and EL-HIIT, no difference was found in peak values (*p* > 0.05) during the session (excluding baseline, warm-up, and recovery). EL-HIIT presented a more pronounced cardiopulmonary and subjective response than HIIT.

## 1. Introduction

Researchers recommend regular exercises that involve improving cardiorespiratory fitness and the muscular system functional capacity, as they can result in a better cardiovascular and metabolic profile [1,2,3]. Concurrent training consists of exercise using aerobic and strength exercises combined in the same session, and it has emerged to maximize training adaptations or health benefits such as improved cardiorespiratory capacity and muscle strength. However, the effect of concurrent training may end up negatively affecting training adaptations such as attenuation of hypertrophy, strength and power when compared to an isolated session of strength training. [4]. In contrast, it appears that high-intensity interval training (HIIT) and sprint interval training (SIT) sessions can be an alternative to minimize the negative effects of concurrent training, achieving better benefits [4].

High-intensity interval training (HIIT) is characterized by repeated efforts close to maximum, at intensities ranging from 80–95% of maximum heart rate [5] or 80–90% of $\overset{\cdot }{V}O_{\mathrm{2max}}$ [6], and has been used as an alternative to concurrent training, demanding less training time being effective in promoting similar or superior health benefits to traditional continuous aerobic exercise [7,8,9,10]. Although HIIT is an effective method for promoting health/performance benefits, the achievement of benefits will depend on the quality and quantity of the training stimulus, i.e., the type of protocol used [11,12]. Several types of HIIT have come to be highlighted, including different ways to reduce reliance on the ergometer and combine aerobic and muscle strength stimuli, such as high-intensity functional training (HIFT), high intensity interval circuit training (HIICT) [11], HIIT bodywork [13], high intensity interval resistance training (HIIRT) [11]. However, the results are still inconclusive regarding the effectiveness of these protocols in the responses of muscle strength with cardiopulmonary capabilities [11]. In addition, most protocols use stimuli with an “all out” intensity, which makes them difficult for the population to perform, requiring a significant stimulus and generating very high fatigue, which ends up being impractical for some individuals due to their age, health, and psychological aspects [14,15].

Currently, several programs have applied high-intensity sessions in sports centers, fitness clubs, and outdoors, and often seek to conduct group classes [16]. Thus, being able to conduct a type of HIIT that does not require special equipment, does not need to reach maximum or “all out” effort, can be performed in a small space, and can be applied on an individual or group basis, would be extremely beneficial for the general community’s adherence to exercise and to health and sports professionals. An example of this type of HIIT is using an elastic resistance [17], a simple and inexpensive tool that is commonly used for physical training in aging individuals [17,18] and athletes [19].

For example, Gasparini-Neto and collaborators demonstrated that a single session of interval exercise with elastic resistance, which consisted of performing ten bouts 1:1 min with fast walking or running back and forth, was clinically effective in reducing blood pressure and blood glucose in elderly hypertensive women [17]. However, the exercise prescription proposed by Gasparini-Neto and collaborators was monitored only by the subjective perception of the effort scale [17]. Considering the need for movement specificity to prescribe exercise with elastic resistance, these same authors proposed an incremental test with reproducible elastic resistance (CPxEL) to determine maximal oxygen consumption and ventilatory thresholds in healthy young individuals [20]. Based on the CPxEL, it will be possible to prescribe the intensity more accurately for this exercise type.

Thus, a specific and individualized test (CPxEL) may allow greater precision in prescribing high-intensity interval exercise using elastic resistance (EL-HIIT), favoring the session load control. This type of exercise and its acute responses are lacking in the literature. Consequently, the acute cardiopulmonary and subjective responses to an EL-HIIT session are still unknown and need clarification to allow health and sport professionals to apply this protocol for their clients safely and with awareness of the responses that may occur. Therefore, this study aimed to describe and compare the cardiopulmonary and subjective responses during an EL-HIIT session and a traditional HIIT session.

## 2. Materials and Methods

### 2.1. Subjects

The procedures were approved by the Federal University of Espírito Santo Ethics Committee on Human Research under the CAAE No 09109319.2.0000.5542 protocol. It was conducted following the principles established in the Declaration of Helsinki, with verbal and written consent obtained from all participants. Initially, 24 participants were recruited, and two dropouts for personal reasons occurred during the experiments. Twenty-two healthy adults, 11 males and 11 females (Table 1), participated in all experiments. For this sample, the statistical power was 87% (two tails *t*-test; effect size moderate and alpha: 5%). The inclusion criteria were BMI (≥18 and ≤25 kg·m^−2^), age between 18 and 35 years, and physically independent (physical activity ≥150 min/week). The exclusion criteria included cardiac, pulmonary, or inflammatory diseases, other medical contraindications, dietary supplements or anabolic steroids, and suspected respiratory tract infections (e.g., COVID-19).

### 2.2. Study Design

The participants were tested four times in 4 weeks during the morning (7:00 and 12:00 a.m.). Each session was separated by seven days. In the first week: The pre-examination included a medical history, physical activity questionnaire, and anthropometric measurements. Then, the participants performed an incremental cardiopulmonary exercise test (CPx) until exhaustion to assess the maximal oxygen consumption and prescribing exercise intensity for the HIIT session on a treadmill. In the second week: All participants performed an incremental cardiopulmonary exercise test with elastic resistance (CPxEL) until exhaustion to assess the maximal oxygen consumption and prescribing exercise intensity for the elastic resistance high-intensity interval training (EL-HIIT) session.

Subsequently, all participants randomly performed two exercise interventions (HIIT and EL-HIIT) on the same day period in the third and fourth week. Intervention intensities were selected according to the specific tests. The workload was based on the results of the $\overset{\cdot }{V}O_{\mathrm{2max}}$ during the CPx or CPxEL and matched for the exercise duration (20 min, excluding warm-up and recovery). CPx and HIIT were performed on a motorized treadmill (Inbrasport Super ATL, Porto Alegre, Brazil) with a fixed 1% slope. CPxEL and EL-HIIT were performed on a rubberized mat with an adjustable belt attached to a 2 m elastic tube (Thera Band^®^, Akron, OH, USA) [20]. All participants were instructed to have a light meal at least 2 h before the test and avoid strenuous exercise, caffeinated products, and alcoholic beverages 24 h before the tests and exercise sessions.

### 2.3. Cardiopulmonary Exercise Test (CPx)

The test was performed on a motorized treadmill maintained at a 1% incline with a 3 min warm-up walking at 4 km·h^−1^ followed by an incremental running protocol. Increments of 1 km·h^−1^ every min until exhaustion were applied. We aimed for protocol durations between 6 and 12 min [21], and verbal encouragement was used at the end of the test. The criteria to identify the test as the maximum was to accept at least three of the following criteria: (a) exhaustion; (b) reaching at least 90% of the maximum heart rate predicted by the formula (220—age); (c) respiratory exchange ratio above 1.05 [22]; (d) peak lactate ≥ 8.0 mmol·L^−1^ [23]; (e) uncoordinated movements.

### 2.4. Cardiopulmonary Exercise Test with Elastic Resistance (CPxEL)

CPxEL was carried out on a rubberized mat with 4.5 m of length demarcated with 11 lines (0 to 10) painted in different colors (white and black) and separated by 30 cm. An adjustable belt with a reinforced closure was initially used, coupled to a 2 m elastic silver tube (Thera Band^®^, Akron, OH, USA) (Figure 1). The elastic tube was changed every two weeks or every 2 cm (1%) increase from its original size. In addition, the elastic was tensioned 361 cm in the last stage. As a result, there was no tearing and no accident.

Initially, the subject performed a brief familiarization with the protocol. Then, the belt, the silicone face mask for gas collection, and the heart rate sensor T31 coded™ (Polar Electro Oy, Kempele, Finland) were adjusted. The CPxEL consisted of alternating steps back and forth against an elastic resistance attached to a belt. The subject performed steps alternating the feet forward and backward with the back-and-forth movement. At each stage change, the foot that initiated the movement was alternated. After 3 min of warm-up (S0), a protocol consisting of increments of one stage (60 cm) per minute following a cadence of 200 bpm (beats per minute) on an eight-stage rubber mat was realized. The subjects were encouraged to follow a rhythm of 180 bpm (~90 steps/min) during warm-up, and 200 bpm (~100 steps/min) during the stages emitted by a metronome app (Cifraclub^®^, Belo Horizonte, Brazil) plugged into a speaker. Pilot testing was performed to determine this cadence with three different cadences (150, 180, and 200 bpm) in five subjects not included in this study, optimizing the protocol. In addition, the elastic was attached parallel to the ground in a load cell (200 kg; EMG System of Brazil, SP Ltd., São Paulo, Brazil). Constant verbal encouragement was applied to maintain the rhythm during stages. If the subject reached the last stage or, if not possible, increment to another stage, 10 (ten) bpm was added every minute until exhaustion. This protocol was the same as described by Gasparini-Neto and collaborators, and more details can be found there [20].

### 2.5. High-intensity Interval Training (HIIT)

Initially, the participants rested for 5 min, and a blood sample was collected from the earlobe at baseline and after an exercise session. The HIIT session started with a 3 min warm-up at 4 km·h^−1^. Subsequently, 10 × 1 min (work): 1 min (rest) at ~85% $\overset{\cdot }{V}O_{\mathrm{2max}}$ in CPx were performed, and passive resting periods [24]. The total exercise time was 20 min (excluding warm-up and recovery) (Figure 2).

### 2.6. High-intensity Interval Training with Elastic Resistance (EL-HIIT)

Initially, the participants rested for 5 min, and a blood sample was collected from the earlobe at baseline and after an exercise session. The HIIT session started with a 3 min warm-up at S0 with a cadence of 180 bpm. Subsequently, 10 × 1 min (work): 1 min (rest) at ~85% $\overset{\cdot }{V}O_{\mathrm{2max}}$ with passive resting periods (10 × 1:1) with a cadence of 200 bpm [24]. The bouts were conducted as follows: fast forward and backward running (from the mat’s beginning to the selected stage’s upper band), where a belt attached the participants with a silver elastic tube (^®^Thera-band Tubing) adjusted at the height of the iliac crest. The participants were verbally encouraged during the effort. The total exercise time was 20 min (excluding warm-up and recovery) (Figure 2).

### 2.7. Measurements

All procedures were performed at a laboratory with a controlled temperature (21–24 °C). The heart rate sensor T31 coded™ (Polar^®^ Electro Oy, Kempele, Finland) was used. Heart rate (HR), maximum oxygen consumption ($\overset{\cdot }{V}O_{\mathrm{2max}}$), minute ventilation ($\overset{\cdot }{V}E$), respiratory exchange ratio (RER), and carbon dioxide output ($\overset{\cdot }{V}CO_{2}$) (measured breath-by-breath by Cortex Metalyzer 3B, Germany; analyzed by Metasoft™) were monitored continuously at rest and during the tests and training sessions. These values were collected at rest (after 5 min of passive rest) and continuously averaged over 10 s intervals. In addition, average and peak values of HIIT and EL-HIIT sessions during the exercise and passive rest were calculated (average: the last 10 s for each bout added to the last 10 s for each passive rest between bouts; peak: last 10 s for each bout, only). For better comparability of the pulmonary parameters, the $\overset{\cdot }{V}O_{2}$ and HR of each training session was set in relation to the $\overset{\cdot }{V}O_{\mathrm{2max}}$ and HR_max_ of the highest value achieved in the tests and depicted as a percentage (%$\overset{\cdot }{V}O_{\mathrm{2max}}$ and %HR_max_).

Blood lactate concentration was analyzed at rest and after each exercise session (HIIT and EL-HIIT) at 3, 5, and 7 min of recovery. In addition, blood samples of 50 µL were taken from the earlobe and analyzed via the electro-enzymatic (YSI 2.300 STAT; Yellow Springs Inc., Yellow Springs, OH, USA).

The BORG-CR10 [25] rate of perceived exertion scale (RPE) for central effort and the OMNI-RES EB scale adapted for elastic resistance with Thera-Band^®^ (Thera Band^®^, Akron, OH, USA) [26] for peripheric effort were applied at baseline, at the end of each bout (HIIT and EL-HIIT), and 10 min post-exercise [27].

To analyze the level of physical activity, the short IPAQ was used, only to see if the participants were physically independent (≥150 min·week^−1^ of physical exercise). Body mass and height were measured using a digital anthropometric scale (Marte Scientific, L200, São Paulo, Brazil), and the Body Mass Index (BMI) was calculated. The Jackson–Pollock 3-site uses chest, abdomen, and thigh skinfold test results for males; and triceps, thigh, and suprailiac skinfold test results for females to calculate body fat (Mitutoyo Cescorf, Porto Alegre, Brazil) [28].

### 2.8. Statistical Analysis

All values are expressed as the means and standard deviation. All data were tabulated and double-verified by independent researchers. The analysis was performed using the SPSS 28.0.1 software. The normality was tested using the Shapiro–Wilk test and submitted to an evaluation of histogram, kurtosis, and skewness. Paired *t*-test was used to compare peak and average values between sessions (HIIT vs. EL-HIIT). Cohen’s *d* effect size from an arbitrary scale was calculated and classified as trivial (0–0.19), small (0.20–0.49), moderate (0.50–0.79), and large (≥0.8) to determine the magnitude of differences [29,30]. Two-way repeated measures ANOVA and mixed model with Bonferroni’s post hoc test were used to compare HIIT and EL-HIIT during the sessions (time vs. protocol; each bout). The Greenhouse–Geisser correction was considered when a lack of sphericity was noted. The significance level was set at *p* < 0.05.

## 3. Results

### 3.1. Cardiopulmonary Exercise Test (CPx) and Cardiopulmonary Exercise Test with Elastic Resistance (CPxEL)

Maximum values of CPx and CPxEL are shown in Table 2. The participants achieved a $\overset{\cdot }{V}O_{\mathrm{2max}}$ at 14.3 ± 1.7 km·h^−1^ in CPx. The maximum stage was achieved at 6.3 ± 1.3, reached a cadence of 205 ± 12 bpm, and a maximal force of 19.5 ± 3.4 kg in CPxEL.

### 3.2. Comparison of Exercise Sessions

The presentation of the two interventions in this study is focused on the peak, the average, and during the entire session (each bout).

#### 3.2.1. Peak Values of the Exercise Sessions (Excluding Rest Intervals)

The average speed was 11.6 ± 1.7 km·h^−1^ for HIIT and 5.9 ± 0.1 km·h^−1^ for EL-HIIT (total distance/time). In EL-HIIT, the average distance covered was 97.41 m (distance until each stage × cadence), and the average cadence was 25.94 times (number of runs back and forth). During the EL-HIIT, the average stage was 3.6 ± 1.5 with a cadence of 200 bpm. As a result, the subjects achieved an average HR_max_ percentage of 84.6 ± 3.4% in HIIT and 90.5 ± 3.6% in EL-HIIT. In addition, they achieved an average $\overset{\cdot }{V}O_{\mathrm{2max}}$ percentage of 78.9 ± 2.3% in HIIT and 86.1 ± 2.5% in EL-HIIT.

EL-HIIT was significantly higher than HIIT for the parameters HR, $\overset{\cdot }{V}E$, relative $\overset{\cdot }{V}O_{2}$, absolute $\overset{\cdot }{V}O_{2}$, $\overset{\cdot }{V}CO_{2}$, BORG-CR10, and OMNI-RES EB (*p* < 0.05; Table 3). There was no difference in peak RER, lactate, and glucose between HIIT and EL-HIIT (*p* > 0.05; Table 3). The effect size (ES) was large for HR, $\overset{\cdot }{V}E$, and lactate (≥0.8), but there was small to moderate ES for relative $\overset{\cdot }{V}O_{2}$, absolute $\overset{\cdot }{V}O_{2}$, $\overset{\cdot }{V}CO_{2}$, RER, BORG-CR10, and OMNI-RES EB (<0.8) (Table 3).

#### 3.2.2. Average of the Exercise Sessions (Peak and Rest Intervals)

EL-HIIT showed a significantly higher $\overset{\cdot }{V}E$ and RER than HIIT (large effect size ≥ 0.8; *p* < 0.05). HR, absolute and relative $\overset{\cdot }{V}O_{2}$, and $\overset{\cdot }{V}CO_{2}$ were also significantly higher (*p* < 0.05), but with a small to moderate ES. No statistical difference was observed for RER, but it showed a large effect size (*p* > 0.05; Table 4).

The average of HR was 76.8 ± 7.8% and 83.7 ± 9.3%, and $\overset{\cdot }{V}O_{2}$ was 57.9 ± 7.71% and 64.7 ± 9.4% for HIIT and EL-HIIT, respectively.

#### 3.2.3. Entire Exercise Session (Baseline, Warm-Up, Each Work: Rest and Recovery)

The HR during HIIT and EL-HIIT increased over time (Figure 3A). During HIIT, the HR reached 85% HR_max_ after the fourth bout and was maintained within the range (85–95% HR_max_) in 70% of bouts. During EL-HIIT, the HR reached the target range (85% HR_max_) after the third bout and was maintained within the target range (85–95% HR_max_) in 80% of the bouts.

The $\overset{\cdot }{V}O_{2}$ increased over time until the fourth bout and stabilized (Figure 3B). During HIIT, the $\overset{\cdot }{V}O_{2}$ reached 80% $\overset{\cdot }{V}O_{\mathrm{2max}}$ after the fourth bout and was maintained within the target range (80–90% $\overset{\cdot }{V}O_{\mathrm{2max}}$) in 70% of the bouts. During EL-HIIT, the $\overset{\cdot }{V}O_{2}$ reached 80% $\overset{\cdot }{V}O_{\mathrm{2max}}$ after the second bout and was maintained within the target range (80–90% $\overset{\cdot }{V}O_{\mathrm{2max}}$) in 90% of the bouts.

The RPE BORG-CR10 increased over time (Figure 3C). During HIIT, the BORG-CR10 reached somewhat hard (number 4) after the seventh bout and reached a maximum of 4.6 on the BORG-CR10 scale. During EL-HIIT, the BORG-CR10 reached somewhat hard (number 4) after the fifth bout and reached a maximum of 5.7 on the BORG-CR10 scale.

The RPE OMNI-RES EB increased over time until the eighth bout and stabilized (Figure 3D). During HIIT, the OMNI-RES EB reached number 4 after the sixth bout and reached a maximum of 5 on the OMNI-RES EB scale. During EL-HIIT, the OMNI-RES EB reached number 4 after the third bout and reached a maximum of 6.2 on the OMNI-RES EB scale.

At baseline, no statistical differences were found in physiological and subjective parameters (*p* > 0.05). Figure 3 shows the time course (baseline, warm-up, work:rest, and recovery) of HR, $\overset{\cdot }{V}O_{2}$, BORG-CR10, and OMNI-RES EB across the HIIT and EL-HIIT. The HR and $\overset{\cdot }{V}O_{2}$ at warm-up of EL-HIIT were significantly higher than HIIT (*p* < 0.0001; Figure 3A). No significant difference was found for HR during the session and recovery (*p* > 0.05). The rest between bouts 6 and 7 (*p* = 0.029) and after 5min recovery (R5; *p* = 0.037) were higher statistically for EL-HIIT than HIIT (Figure 3B).

Figure 3C,D show the course of the RPE. No significant difference was observed for BORG-CR10 (*p* > 0.05) (warm-up, each work: rest, and recovery). OMNI-RES EB presented statistical differences only for the warm-up for EL-HIIT (*p* = 0.007).

## 4. Discussion

The present study compares cardiopulmonary and subjective responses during EL-HIIT with a traditional HIIT session. The main findings of this study were: (1) EL-HIIT exerts greater acute physiological and subjective effects compared to HIIT in peak values, and (2) there was a similar response during each bout of EL-HIIT and HIIT. As far as we know, this is the first study that describes the acute cardiopulmonary response to a HIIT session using an elastic resistance (EL-HIIT) and compares it to treadmill HIIT. The load control during HIIT performance is challenging for coaches and practitioners without ergometers. However, this HIIT protocol using elastic gives us the opportunity to control the load using a specific test before prescribing the session at the targeted intensity at a low cost.

The difference between the HIIT and EL-HIIT sessions does not invalidate the new EL-HIIT protocol; on the contrary, it demonstrates precise prescription and the reach of responses in physiological parameters through a new and specific test [19]. Furthermore, the comparison with a treadmill HIIT was proposed to verify that the acute responses found in the EL-HIIT would not be inferior to a HIIT session already well consolidated in the literature [5,15,31,32].

### 4.1. Peak and Average Values of Exercise Session

Although the peak session values (excluding rest between bouts) of EL-HIIT (90.5 ± 3.6%) and HIIT (84.6 ± 3.4%) in the present study are different, both are close to the average presented by Costa and collaborators [31]. The authors obtained an average of 87.4 ± 4.0% of the HR_max_ of a HIIT session with passive rest interval [31]. On the other hand, Falz and collaborators presented an average of peak values in $\overset{\cdot }{V}O_{\mathrm{2max}}$ of 98%, which is far above what was prescribed and the result demonstrated in the present study [33]. However, the authors prescribed 85–95% of HR_max_ on the cycle ergometer, used the 4 min protocol, and the rest interval between bouts was active, unlike the rest intervals in the present study, which were passive, interfering with the intensity of the session [33].

Furthermore, when comparing the average session values (peak rest between bouts), the present study found a mean of about 76.8 ± 7.8% of HR_max_ in HIIT and 83.7 ± 9.3% of HR_max_ in EL-HIIT. These results are also close to studies that demonstrated HR_max_ percentages of 81.4 ± 4.1% for HIIT 1:1 min [31], 79.7 ± 5.0% for HIIT 30:30 s, and 80.9 ± 4.5% for HIIT 3 min [34]. Another essential variable is the $\overset{\cdot }{V}O_{2}$, that during a HIIT session, the mean percentual of $\overset{\cdot }{V}O_{\mathrm{2max}}$ was about 59–66% (59.7 ± 3.7 for HIIT 30s and 66.0 ± 6.6 for HIIT 3 min) [34]. These results were also close to the present study with an average $\overset{\cdot }{V}O_{\mathrm{2max}}$ of 57.9 ± 7.71% in HIIT and 64.7 ± 9.4% in EL-HIIT. Although EL-HIIT showed closer values than HIIT, both protocols were considered high intensity since they reached at least ~85% of HR_max_ and >60% of $\overset{\cdot }{V}O_{\mathrm{2peak}}$ [5,15]. These findings demonstrated that the CPxEL had a more precise response for the prescription of the EL-HIIT session, which may be due to the difference in ergometer since EL-HIIT does not use an ergometer, despite using a protocol with elastic resistance [17,20]. Therefore, it is interesting to note that the session specificity with elastic resistance (EL-HIIT) differs from running on a treadmill in that it will have a forward and backward running movement with elastic resistance, which involves greater eccentric contraction during the session. Some evidence point out that eccentric contraction can aid in rapid gains in strength, muscle mass, mobility, and independence, and can prevent and reduce sarcopenia and the risk of falls [35]. Furthermore, evidence suggests that elastic resistance can promote hypertrophy, strength, and power. Therefore, its benefits have been seen not only in rehabilitation but for healthy individuals and athletes [36,37].

In this sense, although the sessions in the present study had the same duration and intensity for $\overset{\cdot }{V}O_{2}$, the response to the sessions differed in physiological responses. This is expected since some authors emphasize that for high percentages of $\overset{\cdot }{V}O_{\mathrm{2max}}$, there may be greater heterogeneity responses between individuals [38]. Therefore, this shows that both HIIT and EL-HIIT can be used as training protocols. However, each has its particularities (different participation in concentric/eccentric and aerobic strength stimuli) and achieves the desired percentage at different times.

On the other hand, the present study showed a peak RPE BORG and OMNI RES EB response between 3 and 5 (moderate–hard), which was lower when compared to other studies in both sessions that found a mean of 6.63 ± 1.35 [32] and above 7 [39] in perceived exertion, although the physiological response of EL-HIIT was equal or higher to that found in the literature. This difference may be a result of the type of ergometer, as the literature mainly used cycle ergometers [32] and active rest interval [39], which may interfere with the peak perception of the session, as the cycle ergometer will recruit more specific muscles. Furthermore, EL-HIIT obtained higher cardiopulmonary and subjective responses than HIIT, which may be precise due to greater muscle strength generated by using elastic resistance and greater eccentric participation [37], thus elevating the acute responses in EL-HIIT.

### 4.2. Entire Exercise Session (Baseline, Warm-Up, Each Work: Rest, and Recovery)

Few studies show how long the subjects maintained in the targeted HR and $\overset{\cdot }{V}O_{2}$ range, which is critical information since training effects change with duration, intensity, frequency [40], and protocols/ergometer [41]. In the present study, subjects needed around six to seven minutes to adjust the cardiopulmonary load during HIIT and three to four minutes during EL-HIIT (excluding warm-up). Furthermore, it is normal in HIIT protocols to have about 40 percent of the session below the target training zone [33]. Therefore, it can be assumed that although the prescribed intensity was the same for both, each protocol had a different response. The acute physiological responses of EL-HIIT overestimated the prescribed percentage intensity value, reaching the target range faster. At the same time, HIIT underestimated the prescribed value, taking longer to reach the target range, although both were within the prescribed intensity range. This difference in the acute response may impact the post-exercise and chronic effects, but further studies are needed. To date, we found only one chronic study that involved a training session close to the present study’s proposal (protocol 1–2 series of 12–14 4 s shots with 26 s rest interval). However, the author’s objective was to investigate the effect of sprint training with an elastic resistance exclusively on anteroposterior force production, concluding that sprint training with an elastic was able to increase the capacity to produce force in soccer athletes [37]. These findings reinforce the importance of the present study, bringing as a novel the session protocol and the physiological and subjective responses that mix aerobic and muscular strength stimuli with the use of elastic resistance. This may help in training athletes [37] and individuals with and without comorbidities.

On the other hand, when analyzing studies that tracked the perceived exertion scale of each bout in a 10 × 1:1 min protocol [32,39] also showed higher perceptions than what was found in the present study. However, it is known that a session on a cycle ergometer can have higher RPE than a session on a treadmill [42]. One of the studies cited previously used a cycle ergometer [32], which may be a factor that explains having a higher RPE than the present study that used a treadmill and an elastic band. The other study used RPE as a prescription parameter, but the session was adjusted to achieve the desired RPE [39]. Both studies employed different variables (power, HR, and RPE) to prescribe the exercise session [32,39]. This was different from the present study that used a fixed load of the percentage of $\overset{\cdot }{V}O_{\mathrm{2max}}$ and also performed a specific pre-test for each session to prescribe correctly (CPx and CPxEL).

These factors (different ergometer and prescription variables) may have influenced the perception of the RPE scales even though EL-HIIT has an acute cardiopulmonary physiological response equal to or greater than the literature. Therefore, although the present study presented in detail the cardiopulmonary and subjective responses during an EL-HIIT session, it would be essential to have studies demonstrating these responses in post-exercise recovery and its chronic effects to verify its possible health benefits.

### 4.3. Limitations

Some limitations need to be mentioned. First, despite instructions for subjects to abstain from vigorous physical activity, caffeinated products, and alcoholic beverages 24 h before EL-HIIT and HIIT sessions, and to maintain a good sleep pattern, we did not evaluate these aspects. Second, this study was applied only to healthy individuals, suggesting this protocol’s application in different populations with different conditions (with comorbidities, for example).

## 5. Conclusions

A single session of high intensity interval training with elastic resistance (EL-HIIT) presented a more pronounced cardiopulmonary and subjective response than HIIT. These findings point to the need to study different populations to evaluate acute and chronic effects of this new HIIT proposal, considering its potential to develop combined cardi-orespiratory fitness and muscle strength.

## Figures and Tables

**Figure 1 ijerph-20-06061-f001:**
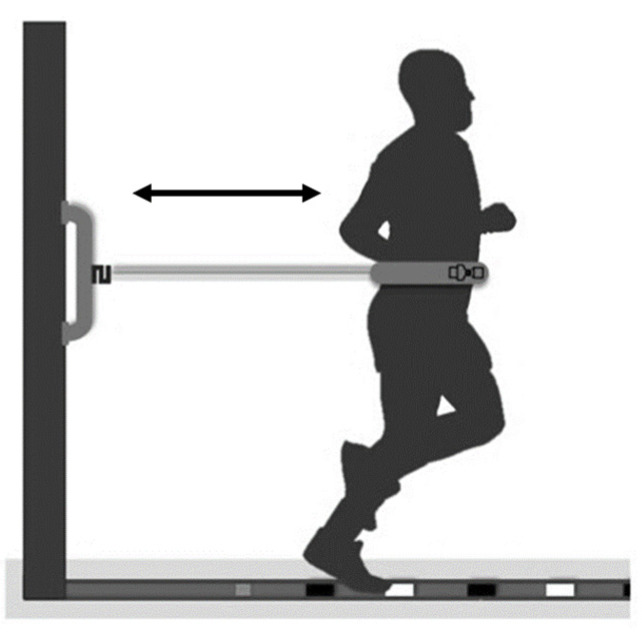
Graphical representation of a participant doing the EL-HIIT on a rubberized mat. During the exercise, the subject runs to the line determined by the CPxEL and back to the beginning of the mat repeatedly for one minute with a one-minute passive rest interval (10 × 1:1 min).

**Figure 2 ijerph-20-06061-f002:**
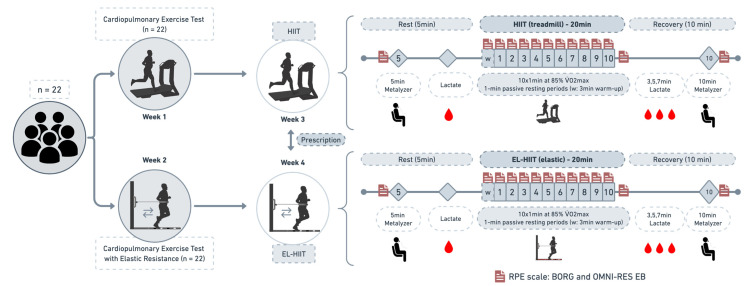
Study design. The first day was set as a cardiopulmonary exercise test (CPx) to examine HR_max_ and $\overset{\cdot }{V}O_{\mathrm{2max}}$. The second day a cardiopulmonary exercise test with elastic resistance (CPxEL) to examine HR_max_ and $\overset{\cdot }{V}O_{\mathrm{2max}}$. An acute HIIT and EL-HIIT bout were randomly performed in the third and fourth weeks. Physiological and subjective parameters were measured before (pre) the interventions, during, and 10 min after stopping the exercise. RPE: rate of perceived exertion. Metalyzer: gas analysis.

**Figure 3 ijerph-20-06061-f003:**
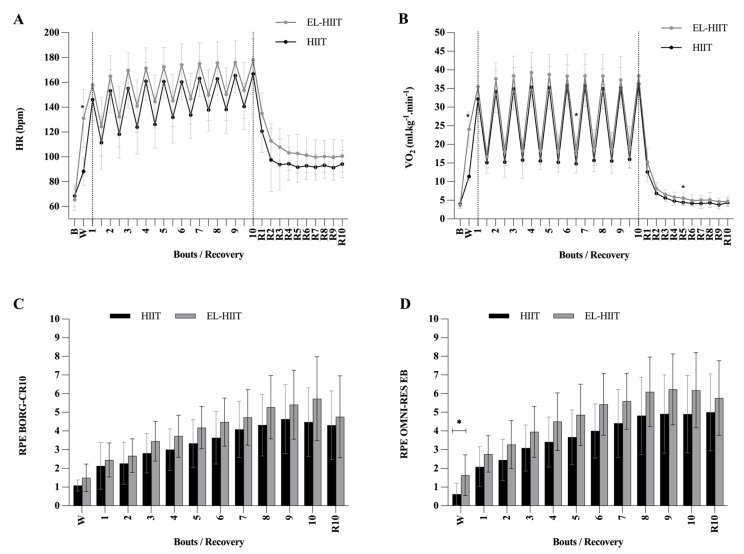
Graphs show the peak physiological and subjective responses during HIIT and EL-HIIT with baseline (B), warm-up (W), and recovery periods (R). Mean ± SD. HIIT: high-intensity interval training; EL-HIIT: high-intensity interval training with elastic resistance; HR: heart rate (**A**); $\overset{\cdot }{V}O_{2}$: oxygen uptake (**B**); BORG-CR10 (**C**) and OMNI-RES EB (**D**): rate of perceived exertion scales representing central and peripheric effort, respectively. R1–R10: 10 min recovery, * HIIT vs. EL-HIIT (*p* < 0.05).

**Table 1 ijerph-20-06061-t001:** Baseline characteristics measured during passive rest (n = 22).

Age and Anthropometric Parameters	
Age (years)	27.6 ± 4.4
Height (cm)	1.71 ± 0.10
Body Mass (kg)	66.9 ± 10.6
BMI (kg·m^−2^)	22.8 ± 2.3
Body Fat (%)	14.5 ± 8.7
**Physiological Parameters**	
HR (bpm)	69 ± 9
$\overset{\cdot }{V}E$ (L·min^−1^)	9.75 ± 3.16
$\overset{\cdot }{V}O_{2}$ (mL·kg^−1^·min^−1^)	4.07 ± 1.27
$\overset{\cdot }{V}O_{2}$ (L)	0.27 ± 0.10
$\overset{\cdot }{V}CO_{2}$ (L)	0.22 ± 0.08
RER	0.79 ± 0.06
Lactate (mmol·L^−1^)	0.97 ± 0.42

Values are presented as the means and standard deviation. BMI: body mass index; HR: heart rate; $\overset{\cdot }{V}E$: pulmonary ventilation; $\overset{\cdot }{V}O_{2}$: oxygen uptake; $\overset{\cdot }{V}CO_{2}$: carbon dioxide output; RER: respiratory exchange ratio; Lactate: blood lactate concentration.

**Table 2 ijerph-20-06061-t002:** Maximum values in CPx and CPxEL.

Physiological Parameters	CPx	CPxEL
HR_max_ (bpm)	189 ± 9	182 ± 12
$\overset{\cdot }{V}E_{\max}$ (L·min^−1^)	106.10 ± 23.25	112.46 ± 24.43
$\overset{\cdot }{V}O_{\mathrm{2max}}$ (mL·kg·min^−1^)	44.35 ± 5.83	40.86 ± 4.21
$\overset{\cdot }{V}O_{\mathrm{2max}}$ (L)	2.97 ± 0.75	2.76 ± 0.59
$\overset{\cdot }{V}CO_{\mathrm{2max}}$ (L)	3.14 ± 0.81	2.88 ± 0.62
RER_max_	1.06 ± 0.04	1.05 ± 0.03
Lac_peak_ (mmol·L^−1^)	10.12 ± 3.13	9.14 ± 2.63
**Subjective Parameters**		
BORG-CR10 (0–10)	7.50 ± 1.87	8.45 ± 1.57
OMNI-RES EB (0–10)	7.64 ± 1.68	8.59 ± 1.30

Values are presented as the means and standard deviation. CPx: cardiopulmonary exercise test; CPxEL: cardiopulmonary exercise test with elastic resistance; HR: heart rate; $\overset{\cdot }{V}E$: ventilation; $\overset{\cdot }{V}O_{2}$: oxygen uptake; $\overset{\cdot }{V}CO_{2}$: carbon dioxide output; RER: respiratory exchange ratio; Lac: blood lactate concentration; BORG-CR10 and OMNI-RES EB: perceived exertion scales representing central and peripheric effort, respectively.

**Table 3 ijerph-20-06061-t003:** Peak values in exercise sessions (excluding warm-up and recovery phases).

Physiological Parameters	HIIT	EL-HIIT	*p*	Cohen’s *d* [95% CI]
HR (bpm)	158.79 ± 13.75	171.45 ± 15.93	**0.013 ***	0.85 ^L^ [0.23 to 1.47]
$\overset{\cdot }{V}E_{\max}$ (L·min^−1^)	70.51 ± 18.41	87.73 ± 24.56	**0.000 ***	0.80 ^M^ [0.18 to 1.41]
$\overset{\cdot }{V}O_{\mathrm{2max}}$ (mL·kg·min^−1^)	34.98 ± 5.36	38.02 ± 5.06	**0.000 ***	0.58 ^M^ [−0.02 to 1.19]
$\overset{\cdot }{V}O_{\mathrm{2max}}$ (L)	2.36 ± 0.65	2.56 ± 0.67	**0.000 ***	0.30 ^S^ [−0.29 to 0.90]
$\overset{\cdot }{V}CO_{\mathrm{2max}}$ (L)	1.97 ± 0.57	2.23 ± 0.65	**0.000 ***	0.43 ^S^ [−0.17 to 1.02]
RER	0.83 ± 0.05	0.86 ± 0.04	0.362	0.66 ^M^ [0.06 to 1.27]
Lac (mmol·L^−1^)	3.49 ± 0.92	6.14 ± 2.87	0.656	1.24 ^L^ [0.60 to 1.89]
**Subjective Parameters**				
BORG-CR10 (0–10)	3.47 ± 1.26	4.20 ± 1.17	**0.039 ***	0.60 ^M^ [0.00 to 1.20]
OMNI-RES EB (0–10)	3.78 ± 1.41	4.89 ± 1.44	**0.035 ***	0.78 ^M^ [0.17 to 1.39]

Values are presented as the means and standard deviation. HIIT: high-intensity interval training; EL-HIIT: high-intensity interval training with elastic resistance; HR: heart rate; $\overset{\cdot }{V}E$: ventilation; $\overset{\cdot }{V}O_{2}$: oxygen uptake; $\overset{\cdot }{V}CO_{2}$: carbon dioxide output; RER: respiratory exchange ratio; Lac: blood lactate concentration; BORG-CR10 and OMNI-RES EB: perceived exertion scales representing central and peripheric effort, respectively; CI: confidence interval. ^S^ = Small; ^M^ = Moderate; ^L^ = Large. * (bold) HIIT vs. EL-HIIT (*p* < 0.05).

**Table 4 ijerph-20-06061-t004:** Mean values for the sessions (peak and rest, excluding warm-up and recovery).

Physiological Parameters	HIIT	EL-HIIT	*p*	Cohen’s *d* [95% CI]
HR (bpm)	145 ± 15	158 ± 18	**0.002 ***	0.78 ^M^ [0.17 to 1.40]
$\overset{\cdot }{V}E_{\max}$ (L·min^−1^)	55.94 ± 13.29	70.99 ± 19.81	**0.001 ***	0.89 ^L^ [0.27 to 1.51]
$\overset{\cdot }{V}O_{\mathrm{2max}}$ (mL·kg·min^−1^)	25.72 ± 3.42	28.70 ± 4.16	**0.002 ***	0.21 ^S^ [−0.38 to 0.80]
$\overset{\cdot }{V}O_{\mathrm{2max}}$ (L)	1.73 ± 0.44	1.93 ± 0.53	**0.000 ***	0.41 ^S^ [−0.19 to 1.01]
$\overset{\cdot }{V}CO_{\mathrm{2max}}$ (L)	1.53 ± 0.41	1.83 ± 0.55	**0.000 ***	0.62 ^M^ [0.01 to 1.22]
RER	0.92 ± 0.05	0.98 ± 0.05	0.166	1.20 ^L^ [0.56 to 1.84]

Values are presented as the means and standard deviation. HIIT: high-intensity interval training; EL-HIIT: high-intensity interval training with elastic resistance; HR: heart rate; $\overset{\cdot }{V}E$: ventilation; $\overset{\cdot }{V}O_{2}$: oxygen uptake; $\overset{\cdot }{V}CO_{2}$: carbon dioxide output; RER: respiratory exchange ratio; Lac: blood lactate concentration; BORG-CR10 and OMNI-RES EB: perceived exertion scales representing central and peripheric effort, respectively. ^S^ = Small; ^M^ = Moderate; ^L^ = Large. * (bold) HIIT vs. EL-HIIT (*p* < 0.05).

## Data Availability

Not applicable.

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
