# Peer review of "Acute Cardiopulmonary Response of High-Intensity Interval Training with Elastic Resistance vs. High-Intensity Interval Training on a Treadmill in Healthy Adults"

_ijerph, 2023, doi:10.3390/ijerph20126061_

Round 1

Reviewer 1 Report

The current research examined if acute cardiopulmonary response of HIIT in treamill and with elastic resistance would differ.  I read the paper line by line and I congraculate the authors wrote a nice paper. Method is so nice but I think introduction of the paper should be re-written and readed again and again by all authors to be more precise and explain why this study conducted and which gaps this study fills.

Line 31-32: in this sentence, concurrent training putted forward, however, in the next sentence it is stated that concurrent training may negatively affected training adaptations which is no explained how this occur ?

Line 39: citation format is with flaw, should be in numberic.

Line 49-51: I am not agree with you about “load”. Why you define “all out” trainings as difficult to control the load of training ?

Line 52-58: what is a need to this paragrapgh ? it can be erased.

Line 59-64: so complicated and not understandable.

Flow of introduction explaning why this study conducted and what cotributes to the literature and supporting with evidince is so weak. Whole intro should be rewritten with carefully, otherwise reader cannot be convinced if the current study is important to literature.

Methods:

Physiological parameters in table 1 measured during passive rest ?

Line 101: participants attended the test seesions fasted or fed ? if fed, how you control the diet regimen ?

Line 103: which questionnaire did you used to measure physical activity ? how you mesure anthropometric parameters ? via which device ?

Line 157: erase “passive resting periods”, you already wrote it in the same sentence, whole paper needs to be carefully readed again by all authors.

Line 204: No need to state how figures made, should erase “graphpad……”.

All P values shoul be written to the text during writing the results.

In the discussion

Line 304: is there differences betwen 2 hit protocol or not ? results 1 and 2 seems weird ?

As a whole discussion is enough and succesful.

Future study suggestion should not be in “limitations” session.     

Author Response

We thank you for all your consideration, as they were all very helpful in improving our manuscript. All changes/modifications were marked directly in the text using the "track change" tool. Please see the attachment to see the response.

Reviewer 2 Report

This study aimed to describe and compare cardiorespiratory and subjective responses during high-intensity interval training with elastic resistance (EL-HIIT) and traditional high-intensity interval training (HIIT) sessions. The results of this study demonstrate that interval exercise with elastic resistance (EL-HIIT) is higher in intensity and exhibits more pronounced cardiorespiratory and subjective responses compared to HIIT.

These findings can be fully utilized in sports and healthy exercise instruction. In particular, the fact that this study can be conducted without special equipment is highly beneficial. This study is an important and suggestive study that corresponds to on-site instruction.

Major Comments

In this study, 24 participants were recruited, and two dropped out, resulting in a total of 22 healthy adults.

The sample size is a critical perspective in intervention research. Is there a clear rationale for setting the recruitment of 24 participants during the initial recruitment phase?

The strength setting of the elastic tubing used in the cardiopulmonary exercise stress test with elastic resistance is critical. Is there a rationale for the strength setting of the elastic tubing in this intervention study?

Step exercises with progressively increasing intensity were performed in the cardiopulmonary stress test with elastic resistance. During each stage, verbal encouragement is constantly provided to the subject by those around them to maintain rhythm. Confirmation of the execution of the exercise is a critical point to ensure that the correct load is being applied.

In this case, how was the confirmation of exercise (step exercise) execution (whether it was done or not) carried out? In addition, the transition to other stages may have been conducted depending on the subject's condition. Still, it is essential to clarify how the criteria were used to make such decisions.

Author Response

We thank you for all your consideration, as they were all very helpful in improving our manuscript. All changes/modifications were marked directly in the text using the "track change" tool. Please see the attachment.

Round 2

Reviewer 1 Report

Congrats for the work of authors on the manuscript to get it higher quality, now it is looking better.